# Multi-Scale Drivers of Land-Use Changes at Farm Level I: Conceptual Framework and Application in the Highly Flooded Zone of the Vietnamese Mekong Delta

Thuy Ngan Le [1,2,3,*], Arnold K. Bregt [4] 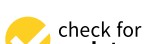, Gerardo E. van Halsema [1,*], Petra J. G. J. Hellegers [1] and Thi Thu Trang Ngo [3]

1 Water Resources Management Group, Wageningen University and Research, P.O. Box 47, 6700 AA Wageningen, The Netherlands
2 Center of Water Management and Climate Change, Institute for Environment and Resources, Vietnam National University-Ho Chi Minh City, No. 1, Marie Curie Street, Linh Trung Ward, Thu Duc District, Hochiminh City 71300, Vietnam
3 Faculty of Geography, University of Social Sciences and Humanities, Vietnam National University-Ho Chi Minh City, 10-12 Dinh Tien Hoang Street, Ben Nghe Ward, District 1, Hochiminh City 71000, Vietnam
4 Laboratory of Geo-Information Science and Remote Sensing, Wageningen University and Research, P.O. Box 47, 6700 AA Wageningen, The Netherlands
* Correspondence: ngan.le@wur.nl (T.N.L.); gerardo.vanhalsema@wur.nl (G.E.v.H.)

**Abstract:** There is an implementation gap between government plans and land-use changes at the local level in the Vietnamese Mekong Delta. This stands in the way of the sustainable development of the delta, especially in the face of environmental degradation, climate change, and water-use conflicts. To narrow the gap between plans and practice, the government needs a better understanding of what drives land-use decisions at the farm level. Our research developed and applied a multi-scale framework to identify the principal drivers of land-use changes at the farm level in the Vietnamese Mekong Delta over the past 40 years. We conducted semi-structured interviews with 31 farmers in the highly flooded zone, then used transcript analysis to quantify the influence of the drivers mentioned by farmers. We found drastic shifts in land uses, predominantly towards rice intensification. Among the 43 change drivers the farmers mentioned, those operating at the regional scale were particularly influential, including the activities of local authorities, neighborhood effects, and the development of water management infrastructure. Market factors have become more prominent in the last two decades, motivating farmers to shift from double to triple rice or to gradually diversify into vegetables. However, agricultural diversification remains limited by the agro-hydrological context, which favors triple rice cropping, as well as household scale factors such as natural and physical assets of the farm, household capital, and labor capacity. The local community also played a key role in land-use change, though with a double-edged effect, both delaying implementation of central government policy and forcing farmers to follow the majority's decision.

**Keywords:** land-use change; driving factor; multi-scale framework; cross-case comparison; floodplain; Mekong Delta; Vietnam

## 1. Introduction

The Vietnamese Mekong Delta (VMD) spans some four million hectares, of which agriculture occupies 63% [1]. Although the extent of agricultural land has remained stable during the past decade, the farming systems in use have shifted at a rate of 15% per year [2].

Land-use changes in the VMD over the past four decades have closely tracked the development of water management infrastructure and related changes in delta hydrological regimes [2,3]. Starting in the 2000s, high dike enclosures were built, alongside a dense network of irrigation and drainage channels across the delta's upper and middle regions.

These prevented crop fields from flooding in the rainy season, enabling triple rice production [4,5]. Meanwhile, sluices were built in the coastal zone to prevent salinity intrusion in the dry season, again to facilitate intensive rice cultivation [4,6]. These adaptations, however, have increasingly led to conflicts, as shrimp farmers require larger quantities of saline water for their ponds, while rice farmers need freshwater. The government has sought to reconcile the needs of freshwater and saline water farms by modifying some sluices and adjusting sluice operation [7].

A fuller exploitation of land and water resources for agricultural development in the VMD has fueled the rapid growth of rice production [8]. However, unexpected environmental impacts have also increasingly emerged, such as riverbank erosion, new inundation areas downstream, water pollution, and land degradation [3,6,9,10]. These effects, combined with the impacts of climate change, sea level rise, and the construction of hydropower dams and reservoirs upstream, have raised increasing concern, particularly regarding the disappearance of the flood season, the growing severity of salinity intrusion, and the droughts, which have become increasingly frequent in the VMD [11–14]. The consequence, in addition to crop losses and diminished productivity, is a growing threat to the long-term sustainable development of the delta [15].

To cope with these challenges, the Vietnamese government initiated policies to set out an agricultural development trajectory favoring higher quality and higher value agricultural products, rather than further volume increases in rice production [15–17]. In particular, Resolution 120 established in 2017 provided specific orientations for sustainable development and climate change adaptation in the VMD [17]. Greater agricultural diversification is foreseen, embracing sustainable brackish and saline aquaculture and fruit production, alongside rice farming. Water resources in the delta would henceforward be managed through flexible water management measures to support "living with floods, inundation, brackish water, and saline water" [17].

However, studies of land-use changes and the implementation of government plans in Vietnam point to gaps between policies and practices at the local level, attributed to the country's centralized policymaking [12,18,19]. Indeed, the realities and motivations of local actors are varied and may differ from central authorities' points of view [20,21]. This gap may stand in the way of sustainable agricultural development in the delta's various regions. To narrow the gap between plans and practice, the government needs to better understand the factors that drive land-use decisions at the farm level, and how government policies and actions can support and shape these.

Land-use changes in the VMD have been studied using a variety of approaches and perspectives. Scientists in the early 1990s observed and described the transformation of farming in the delta from traditional systems to intensive rice cultivation and crop diversification [22,23]. However, this descriptive approach overlooks context-specific factors that drive land-use decisions on individual farms.

Statistical analysis was applied to investigate relations between land uses and key indicators such as climate and socioeconomic variables [8,24]. Such variables have hardly been applied in small-scale studies because they are usually acquired at the provincial or national level. Moreover, statistical studies depend heavily on the sufficiency and transparency of input data, which cannot always be assured at the local level, particularly over a long period [2].

With the development of remote sensing and geographic information system (GIS) techniques in Vietnam in the 2000s, scientists became able to spatially quantify land-use and land-cover changes in more detail over vaster geographical areas [25,26] and identify spatial correlations between land uses, and land-use changes and hydrological factors [2,5,27,28]. However, a spatial approach cannot capture all drivers of change because many drivers, like village and household socioeconomic characteristics, are difficult to observe on the land surface [29].

Social researchers have analyzed the motivations and characteristics of households, and how these might affect land-use changes, based on surveys and interviews [30–33].

These studies sought a deeper understanding of the micro-level drivers that shape on-farm decisions. However, due to their focus on a specific land-use system, they largely bypass the role of higher level drivers and transboundary aspects.

Acknowledging this, a number of authors have integrated spatial and social approaches to develop agent-based models that aim to simulate land-use changes at the farm and regional level [19,34,35]. These models acknowledge that farm-level land-use decisions in the VMD are driven by multiple drivers operational at different scales. While each of these drivers can affect land-use decisions separately, their combined impacts often exhibit greater complexity due to reciprocal interactions, both between the drivers and with regional characteristics [36–39]. These complex interactions between multiple drivers at different scales, alongside limitations in data and hardware capacity, remain key challenges facing integrated modelling today [36]. Moreover, the data used to represent drivers in many land-use models have been criticized as depicting the "wrong scale of social process", and thus confounding empirical observations [40]. Modeling outcomes can be enhanced by analyzing drivers of change and interactions between them, starting at the farm or household level [29,34,41]. However, to conduct such a study we need a theoretical framework that allows us to incorporate the complexity of multi-scale drivers.

In the current study, we developed such a multi-scale theoretical framework, drawing on existing land-use theories and the literature. We then applied the framework in two case studies, representing very different socio-hydrological settings. The first setting was that of a highly flooded, intensive rice production area in the upper VMD. The second concerned a coastal area of the VMD where salinity intrusion was a key factor in land-use dynamics. Our application of the framework in these two contexts had three main aims: (i) to test the validity and rigor of the developed framework, (ii) to gain a deeper understanding of the complex multi-scale drivers of land-use changes at the farm level, and (iii) to explore the potential role of the socio-hydrological setting in shaping these drivers and the resulting land-use changes.

This article introduces the multi-scale drivers theoretical framework and presents its application in the first case study; that is, in the highly flooded upper delta intensive rice zone of the VMD. In a companion article [42] we present the second application, in the salinity intrusion zone, and address the role of socio-hydrological setting with a cross-case analysis.

## 2. The Multi-Scale Drivers Framework

Land-use change is a purposive process whereby the surface cover of land is modified or converted by human activities [29]. Our multi-scale drivers framework for analyzing changes in land covers and land uses [37] recognizes that such change processes occur at three scales: the region (e.g., rural or urban area), the landscape (e.g., village or watershed), and the unit of production (e.g., household or farm). Moreover, at each of these scales, land-use changes are driven by factors in three dimensions: the modalities of land management, biophysical features, and socioeconomic characteristics. Spatially, the modalities of land management operate at the micro scale (i.e., the household or farm level) and directly affect land cover. The biophysical and socioeconomic characteristics can operate at any scale, from global to local, while also influencing land-use decisions at the micro scale. In addition to the multi-scalar aspect, drivers of change within any one of these dimensions can reciprocally affect the others, as well as having cross-over effects on drivers in other dimensions and across space and time.

Because the interactions between change drivers and land uses are heterogeneous, complex, and context-dependent, it is hard to find a framework suitable for all land-use change studies [41]. To develop our multi-scale framework for analyzing land-use changes at the farm level in the VMD, we drew on previous studies combined with knowledge gained from field trips and interviews with farmers in the case study areas. In particular, our framework builds on the micro-level drivers of land-use change by Hettig et al. (2016) [41] and the on-farm decision-making process described by Valbuena et al. (2010) [38].

Hettig et al. (2016) [41] conducted a systematic review of 91 land-use change studies at the household level in tropical regions conducted since 2000. Among these, they distinguished two types of drivers: macroeconomic variables and micro-level drivers. The macroeconomic variables were identified as policies, population growth, and global markets, while the micro-level drivers were specified as institutions, infrastructure, endowments, markets, and technology. Although that study's classification of drivers is clear and sufficient, we propose that 'endowments' be separated from the other micro-level drivers, as endowments are important in determining a household's internal characteristics, while the other micro-level drivers are more externally oriented. Meanwhile, Valbuena et al. (2010) [38] provided a well-structured account of farmers' decision-making on land uses, considering the interactions between a household's ability and its willingness to respond to external drivers.

In addition to combining concepts from the two above studies, we applied grounded theory [43,44] to complete our framework and deeply explore the drivers that really played a role in land-use decisions at the farm level. Grounded theory is an inductive method for incorporating data from interview transcripts into an existing theory or framework. The data acquisition and analysis follow a cyclical process that uses early data analysis to shape the ongoing data collection. Concepts and linkages between different elements in the text are identified by assigning codes through the interview transcript. Typically, three types of coding methods are conducted: open coding, focused coding, and axial coding. Open coding starts the analytic process by assigning codes to participants' words and statements. The focused coding explores specific issues identified in the previous coding phase, while the axial coding identifies key categories and their relationships to others. The process of coding and developing categories is supported by writing memos that are kept continuously and provide a record of thoughts and ideas to the researcher [43]. Hence, our framework included additional drivers structured in accordance with the context of the VMD. For example, our interviews with farmers revealed a variety of drivers related to household ability. We grouped these into five types of assets: 'natural assets', 'physical assets', 'social assets', 'financial assets', and 'human assets'. We based our definitions of these assets on the sustainable livelihoods framework [45].

Figure 1 presents the multi-scale framework we used to investigate drivers of land-use changes at the farm level in the VMD. In line with [24], who also investigated land-use changes in Vietnam, we considered four external drivers at the global or international scale: 'population change', 'global markets', 'policies', and 'climate change'. Population change includes the growth of the global population, as well as changes in national populations and migration. Key pressures related to population change are the imperative to increase food production and goods manufacturing, and the effect of migration in altering demand for certain products. Moreover, the movement of workers to industrial sectors and urban areas impacts labor availability and can thus change the investment climate around agriculture and rural areas. Global markets represent the demand for food, feed, fuel, and fiber. Fluctuations in such demand can strongly impact agricultural production, bringing drastic changes in land-use patterns. Policies at the macro level refer mainly to tariffs and quotas, which are guided by international trade agreements and influence export production. For instance, high global market demand was decisive in generating the boom in shrimp farming in the VMD. However, if a foreign market applied antidumping duties on imported shrimp products, domestic shrimp farms would be quickly hit [46,47]. Similarly, environmental agreements between countries and with international organizations influence national development plans and land-use policies. Lastly, climate change affects national policies as well as the daily practices of farmers [12,15,24,33].



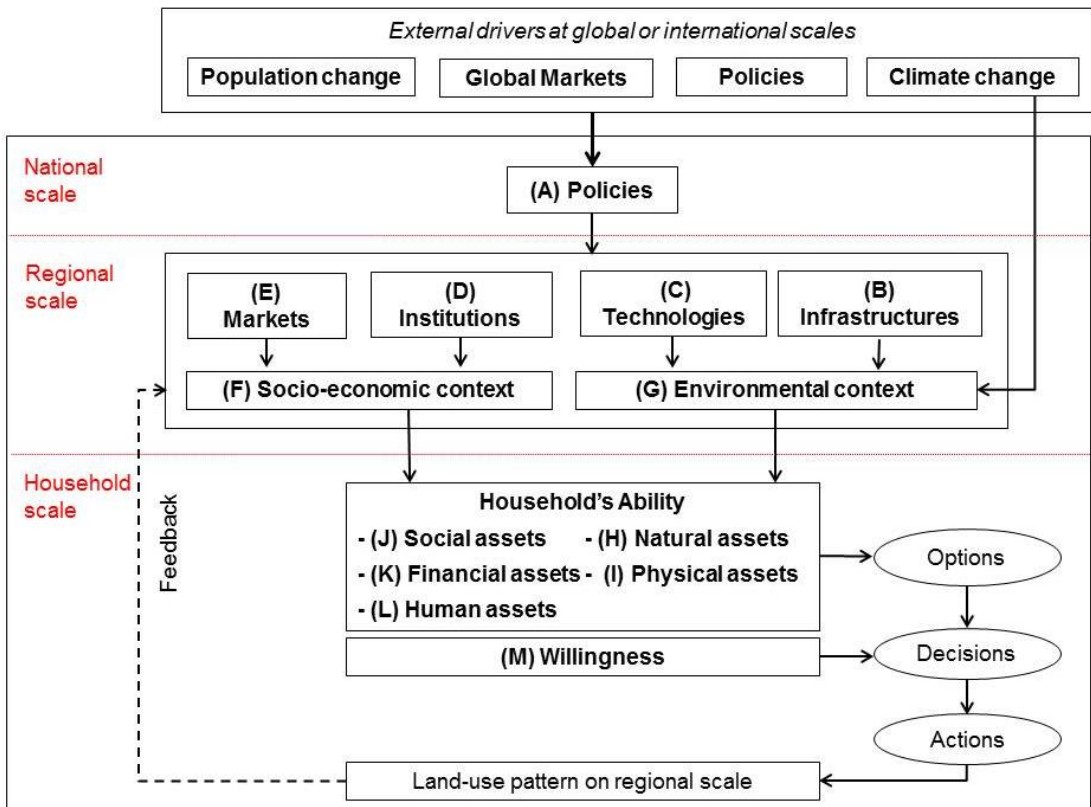

**Figure 1.** Multi-scale drivers framework. The framework builds on the international, national, and regional scales from [41], while the household scale structure is adopted from [38,45]. The letters (A) to (M) represent key factors influencing land-use changes at the farm level.

Though all these external factors undoubtedly impacted the land-use decisions of VMD farmers, they did not come to the fore in our farmer interviews. Instead, the drivers that farmers mentioned coalesced around the national, regional, and household scales.

At the national scale, 'policies' (A) have an overarching role, as they influenced all of the drivers of land-use change mentioned by VMD farmers. We focused particularly on national-scale policies, as these strongly impact all aspects of Vietnamese society [48]. For example, food security has long been a foremost national policy goal in Vietnam. For more than three decades, the country's economy has revolved around intensive rice production. To ensure sustained high annual rice production, the government has put in place a dedicated infrastructure, disseminated new technologies, reinforced domestic markets, and erected local institutions. As a result, from 1986 to 2014 Vietnam's rice production increased dramatically, from 16 to 45 million tons [49]. This greatly exceeded domestic demand and made Vietnam one of the world's top rice exporters. For our research, we referred to a list of key policies presented in a historical review of land and water dynamics in the VMD [2]. These policies include central planning and agricultural collectivization (1975–1986); the *Doi Moi* economic reforms, which brought the decollectivization of land and agricultural production (1986–1988) [18]; market liberalization (since 1989) [50]; policies such as Resolution 9 [51] that allowed provincial governments to replace rice cultivation with other high-value farming activities such as fruit production and shrimp farming [31]; and the recent encouragement of high-value agriculture consistent with a context of climate change [17].

We defined regional drivers of land-use changes as factors external to farms and households, categorized as 'infrastructures' (B), 'technologies' (C), 'institutions' (D), and 'markets' (E). These can relate to different types of region, such as a community, a village, a district, a province or a hydro-agricultural zone or delta. However, because our interviewed farmers did not distinguish the scalar aspect in their accounts and responses,

our data did not enable us to identify the type of region in which the drivers were active. Infrastructures (B) consist of national and provincial roads, electricity and water supply systems, dike rings, sluices, and canal networks. Technologies (C) refers to agricultural machinery, improved varieties, innovative products, and scientific knowledge on farming systems. Institutions (D) were defined as the activities of local governments and officials to implement government policies. Mattner (2004) [49] found that in Vietnam local officials were responsible for disseminating information about national laws and local projects, consulting the community, collecting local inputs, and supervising public works. We defined markets (E) as encompassing the conditions of trade between farmers and small-scale traders or local agricultural agents, including market demand, input and output prices and products, agricultural service availability, and support from private companies. These could be advantageous or disadvantageous to farmers.

To these regional scale drivers, based on Hettigs et al. (2016) [41], we added 'socioeconomic context' (F) and 'environmental context' (G) to the framework after the analysis of the interview transcripts. Socioeconomic context (F) encompasses opinions within the community, neighborhood effects, and changes in local demography (e.g., migration). Environmental context (G) encompasses climate change, technology, and infrastructure, which affected, for example, local weather and flooding regimes, farming schedules, and farmland agro-hydrology (e.g., location of lands inside or outside high dike enclosures).

Markets, institutions, technologies, and infrastructures can affect farmers directly, but their impacts may also be moderated by the socioeconomic and environmental context and changes therein. For instance, information provided by local officials and tradespeople might influence opinions among community members. Community leaders might move early to take advantage of opportunities provided by markets and institutions, thus modeling a switch to a new farming system. Their eventual success can motivate others, since farmers often consider the experiences of their neighbors before deciding to change their own land use [27]. In addition, the VMD socioeconomic context has been marked by the high outmigration of young people, who leave the delta to take advantage of jobs and opportunities in the city. Outmigration has diminished labor availability and thus raised the cost of agricultural production [52]. Meanwhile, the construction of new transportation and water management infrastructures and the use of new technologies have both improved and degraded local environments across the delta. Soil and water quality and flooding regimes have been particularly affected. For example, canal systems have been expanded into remote areas of the VMD in programs to reclaim the highly acid sulfate soils. However, drainage and leaching practices for acid soils have caused the severe acidification of water resources. Also, while rising agrochemical use has facilitated triple rice cropping, it too has contaminated soils and water [3,6]. In addition, the high dikes built to protect farmlands from the annual floods have prevented alluvial deposition in the fields, while also accelerating riverbank erosion. Thus, environmental conditions in the region nowadays exhibit unprecedented change, exacerbated by the simultaneous effects of climate change and upstream hydropower developments in the Mekong basin [13,14].

At the household scale, farmers' decisions and actions are shaped by their 'ability' and 'willingness' to act. Here, ability refers to farm and farmer conditions, which limit land-use options. When farmers want to change their land use, those with strong 'ability' perceive more options and opportunities, while those with weaker ability see more threats [38]. We distinguished five ability-related drivers, drawn from DFID's (1999) [46] sustainable livelihoods guidance sheets: 'natural assets' (H), 'physical assets' (I), 'social assets' (J), 'financial assets' (K), and 'human assets' (L). Natural assets (H) encompass the natural resources available to a farm, including soil and water characteristics and quality, and land shape and topography. Physical assets (I) relate to all accessible infrastructures, agricultural equipment, and lands used for farming. Social assets (J) constitute farmers' networks and relationships within their community and with relatives, neighbors, traders, and officials. Financial assets (K) refer to household incomes, savings, and property, as well as access to loans and government financial support. Human assets (L) reflect demographic

characteristics, such as the age, health, education, and farming experience of the farm household head, alongside the number of children and laborers in the family.

'Willingness' (M) describes farmers' openness to and interest in different land uses. A high enough level of interest can motivate farmers to adjust their 'ability' to switch to a new farming system. In their depiction of the on-farm decision-making process, Valbuena et al. (2010) [38] extended their focus beyond choices made by individual households in order to also consider how change spreads from household to household, accordingly upgrading their model to the regional scale. Hence, when a farmer decides to embark on change, that decision may affect conditions on neighboring farms and the interest of other farmers. Our framework therefore also considers potential feedback effects of land-use changes on higher-level drivers.

## 3. Case Study in the Highly Flooded Upper Delta

### 3.1. Study Area Characteristics

The VMD consists of six agro-hydrological zones, each with its own characteristic topographies, hydrological regimes, and soil types, which shape the prevailing farming systems [2]. Land-use changes across the upper floodplains of the VMD demonstrate the remarkable impact of water interventions on the human-environmental system [3]. The highly flooded zone is composed of the vast natural floodplains of the Long Xuyen Quadrangle in the west and the Plain of Reeds in the east. Located between them is a higher altitude freshwater alluvial zone that offers distinct advantages for agriculture, such as year-round availability of freshwater and good soils (Figure 2). In the highly flooded zone, agriculture used to be limited by soil conditions, which had high acid sulfate levels in the dry season (November to April), and by the rising and falling of floodwaters in the rainy season (May to October). These floodwaters start to rise slowly, by 10–15 cm per day, in July or August, and then peak at 3–4 m in late September or early October [4]. Today, however, intensive rice cultivation in the highly flooded zone is facilitated by a dedicated government effort to control the flood regime and reclaim the acid sulfate soils.

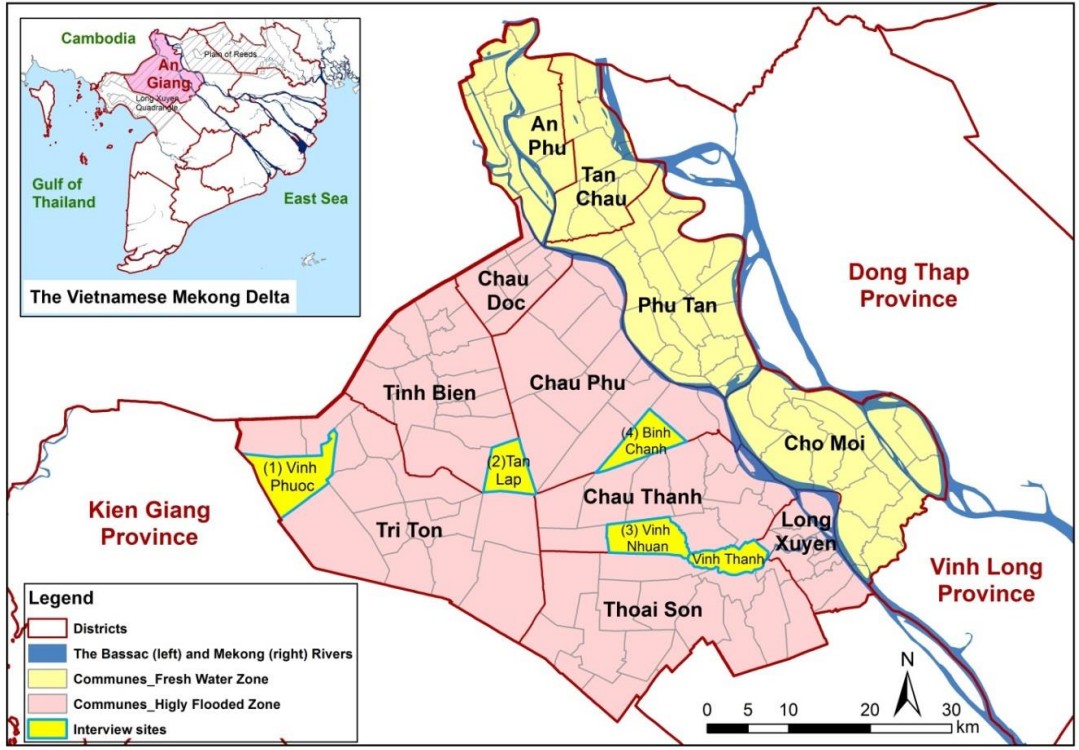

**Figure 2.** Location of research sites in An Giang Province, Vietnamese Mekong Delta.

Central Long Xuyen Quadrangle is part of An Giang Province. Four of the province's districts—Chau Phu, Tinh Bien, Tri Ton and Chau Thanh—form our study area (Figure 2). These districts were chosen because they exhibit the variety of biophysical and socioeconomic characteristics found in the various floodplain regions of the delta. The districts span approximately 1760 square kilometers in total, occupying 50% of the area of An Giang Province. In 2016, 80% of the land in the four districts was devoted to agriculture, with half of that area under triple rice cultivation [53]. The population of the four districts was 673,562 in 2016, comprising 166,643 households [54].

Land uses were similar in the four districts, with more than three quarters of the land under intensive rice. Geographically, the districts can be divided into two zones: east and west. Chau Thanh and Chau Phu districts are located in the east of the studied area, closer to the Bassac River (Figure 2). The topography of these eastern districts is completely flat, while there is a small mountainous area with forest in the western districts. Also, the more eastern districts had a higher percentage of area protected by high dikes (up to 2010) compared to the more western districts [55]. Thanks to their favorable geographies, agricultural conditions, and economies, the districts in the east were more developed than those in the west. Consequently, the poverty rate of Chau Thanh and Chau Phu was lower than in Tinh Bien and Tri Ton [54].

*3.2. Data Collection and Analysis*

To elaborate the role of drivers of land-use changes across spatial and temporal scales, we interviewed VMD farmers to learn their perspectives on land uses and land-use change. To identify interviewees, we used the snowball method, starting with interview subjects recommended by local agricultural officials. The number of farmers interviewed was determined by when we reached saturation of information (Table 1). To obtain information about land-use history and factors influencing decisions to change land uses, we interviewed farmers with relatively long experience in farming and who were active as the main laborer of their household farm. Due to limitations of time and logistical resources, we restricted our interviews to farmers in five communes within the four studied districts. Each commune had a land-use pattern and a history of dike construction considered representative of its district. The communes where interviews were held were Vinh Phuoc (Tri Ton District), Tan Lap (Tinh Bien District), Vinh Nhuan and Vinh Thanh (Chau Thanh District), and Binh Chanh (Chau Phu District) (Figure 2).

First, we were interested in historical farming practices, starting at the time the interviewed farmer obtained the farm. For each farm, we traced an individual transition pathway marking points of change [38,56]. Our interviews were semi-structured and focused on internal and external drivers influencing on-farm decisions [57]. Interview subjects were frequently prompted for deeper explanations, with questions of "why" and "how" certain factors affected their decisions at particular points of change.

After a preliminary analysis of the interview data in January 2016, we revisited the communes in April 2016 for second interviews with the same farmers, or to interview new farmers considered likely to have different land-use change pathways. In total, we conducted interviews in 31 households, and undertook a second interview in 15 households. The interviews were recorded and transcribed. Our transcript analysis first traced the historical development of land use on the farms, alongside events in the households and communities. Since the average age of the interviewees was 50, their memories and experiences in farming started in the 1970s. Figure 3 presents the households' land-use change pathways in a line graph from 1975 to 2016. We chose 1975 as the first year of investigation because that year marked the end of the Vietnam War and ushered in the post-reunification era. This year also marked the start of land reclamation, population settlement, and agricultural expansion in the VMD [2].

**Table 1.** Characteristics of the interviewed farmers and their farms.

| Districts | Tri Ton | Tinh Bien | Chau Thanh | | Chau Phu |
|---|---|---|---|---|---|
| Communes<br>*n:* number of<br>interviewed farmers | Vinh Phuoc<br>(*n* = 6) | Tan Lap<br>(*n* = 9) | Vinh Nhuan<br>(*n* = 6) | Vinh Thanh<br>(*n* = 2) | Binh Chanh<br>(*n* = 8) |
| Age (years)<br>*m:* mean; minimums and<br>maximums in parentheses | *m* = 53<br>(43–65) | *m* = 52<br>(33–74) | *m* = 50<br>(39–65) | *m* = 61<br>(50–71) | *m* = 52<br>(42–72) |
| Number of interviewees with<br>primary school education | 5 | 4 | 3 | 1 | 0 |
| Female household heads | 0 | 1 | 1 | 0 | 0 |
| Immigrants | 4 | 6 | 4 | 0 | 3 |
| Farm size (ha)<br>*m:* mean; minimums and<br>maximums in parentheses | *m* = 8.38<br>(1–33) | *m* = 2.74<br>(0.1–5.6) | *m* = 3.58<br>(2–7) | *m* = 0.49<br>(0.37–0.61) | *m* = 3.73<br>(1.2–9) |
| Number of farms inside high<br>dike enclosures | 3 | 6 | 6 | 2 | 7 |
| Farming systems (number<br>of farms) | | | | | |
|    Triple rice cropping | 2 | 4 | 4 | 1 | 5 |
|    Triple rice + vegetables | 1 | 1 | 1 | 0 | 2 |
|    Double rice cropping | 1 | 2 | 0 | 0 | 0 |
|    Double rice + vegetables | 2 | 1 | 0 | 0 | 1 |
|    Vegetable specialization | 0 | 1 | 1 | 1 | 0 |

　　　　Subsequently, we systematically generated a detailed list of drivers from the empirical data by applying grounded theory with color coding [43,44]. We collected and categorized all drivers that influenced farmers' decisions at each point of change according to the three scales of our framework (Figure 1, Appendix A, Table A1. presents the list of drivers). We discerned key drivers of land-use change decisions by the frequency with which they were mentioned by the interviewed farmers. This frequency was presented as a percentage calculated by counting the number of farmers who mentioned that driver as having played a role in their own land-use change decision.

　　　　We applied the same method of sampling, data acquisition, and analysis in the salinity intrusion zone of the VMD [42]. The use of two such different case study settings enabled us to investigate whether principal drivers differed and whether and how they were shaped by the socio-hydrological context. The findings of this second case study, as well as a cross-case analysis, are presented in a companion article [42].

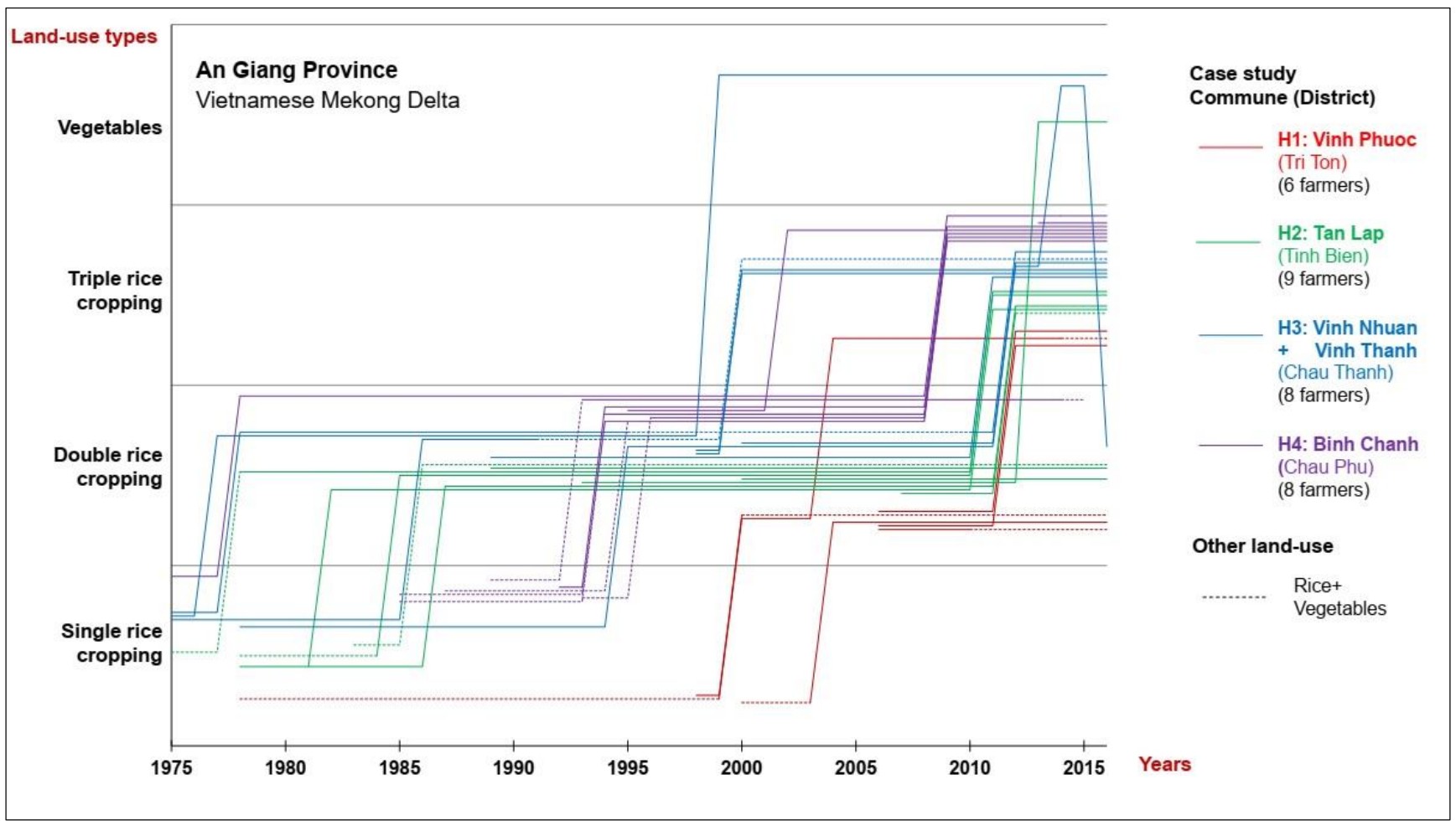

**Figure 3.** Land-use change pathways of 31 households in An Giang Province from 1975 to 2016. The pathways represent land-use changes in the highly flooded zone of the Vietnamese Mekong Delta. Each solid line represents the land-use pathway of an individual farm. The dashed lines represent the mixed farming system with rice and vegetables. Red and green lines represent farms in the western part of the studied area. Blue and purple lines represent farms in the eastern districts.

## 4. Results

### 4.1. Pathways of Land-Use Change in the Highly Flooded Zone

Figure 3 presents pathways of land-use change for the 31 individual farms in the highly flooded zone of An Giang Province. From 1975 to 2016, farmers in the study area used five farming systems: (i) single rice cropping, (ii) double rice cropping, (iii) triple rice cropping, (iv) mixed farming with rice and vegetables, and (v) a vegetable monoculture (though with a variety of types of vegetables). The predominant pattern of land-use change was toward rice intensification. All interviewed farmers started their farming career with rice cultivation and spent many years practicing double rice cropping. Prior to 1975, the war forced farmers off of their lands in the Long Xuyen Quadrangle. In the post-war resettlement era, the new government nationalized these lands and allocated them to others for rice cultivation. At that time, most farmers practiced a single rainfed rice crop annually, because of the high acid sulfate content of the soils and water. To facilitate double rice cropping, the government conducted a large-scale acid sulfate land reclamation program in the early 1980s. They also introduced farmers to high-yield rice varieties that could be harvested in three months. However, the centralized agriculture of the times provided little incentive for high production. With the *Doi Moi* reforms of the late 1980s, land was allocated or sold back to the former owners. Thus, farmers and their families returned and rehabilitated their lands, initiating double rice cropping. In the early 2000s, the construction of high dikes enabled farmers to switch to triple rice cropping. Nearly three quarters of the interviewed farmers had changed to triple rice. Those who kept practicing double rice owned lands outside the high dike enclosures, meaning that their farms were still affected by the annual floods.

Very few of the interviewed farmers had switched from a rice-based farming system to a monoculture of vegetables. Farmers considered such a switch to be high risk, particularly in areas dominated by triple rice cropping. Vegetables require dry roots on high beds. Amid an inundated rice area, these can become settling areas for rodents and other animals. High dikes and a predominance of triple rice in the surroundings also made land preparation for vegetables more difficult and labor intensive. Nonetheless, not all farmers in the highly flooded zone practiced a rice monoculture. From the start there was variation between farmers, with some practicing only single rice cropping and others mixing a single rice crop with vegetables. Most recently, mixed farming with rice and vegetables has begun emerging on farms located outside the high dikes and on farms with elevated beds. These farmers tended to rotate rice and vegetables on the same land, growing rice in the rainy season and vegetables in the dry season. Others used a small plot of land for vegetables, with the larger part reserved for double or triple rice.

The year in which land uses changed varied across the communes, but within any one commune it happened among all farmers almost simultaneously. This was particularly true for the switch from double to triple rice. Farmers said that this change was associated with specific water conditions or a historical event in their community. For instance, rice intensification occurred earlier in the eastern communes (Vinh Nhuan, Vinh Thanh, and Binh Chanh) than in the western communes (Vinh Phuoc and Tan Lap). Farmers could make the switch from a single rice crop to a double rice crop thanks to the availability of innovations such as high-yielding rice varieties, alongside pumping machines and the expansion of canal networks, which occurred from the 1970s to the 1990s. The practice of double rice cropping arrived in Vinh Phuoc last, as there was a cassava plantation here until 2000. When that plantation was dissolved, the land was sold to farmers who then embarked on double rice cropping. The switch to triple rice cropping also happened all at once within a locale. This was due to the construction of high dikes which assured the containment of the annual floods.

### 4.2. Multi-Scale Drivers of Land-Use Change

The predominant land-use change pathway in the highly flooded zone was one of rice-based intensification, reinforced in the past two decades by the expansion of triple

rice cropping. At the regional scale, this trend closely corresponded to the development of infrastructures such as canals to supply freshwater to croplands and dike systems for floodwater control. However, the farmers interviewed mentioned no singular factor as driving their land-use change decisions over time. Instead, a large variety of drivers at the national, regional, and household scales were referenced as being involved in the transitional process of land use. Our transcript analysis identified 43 drivers mentioned by farmers in explaining what led them to decide to change their land use. We grouped these drivers into 13 categories (Figure 4). Regarding the multi-scalar aspect, five policy drivers were identified at the national scale, alongside 23 regional scale drivers and 15 household scale drivers (see Appendix A, Table A1).

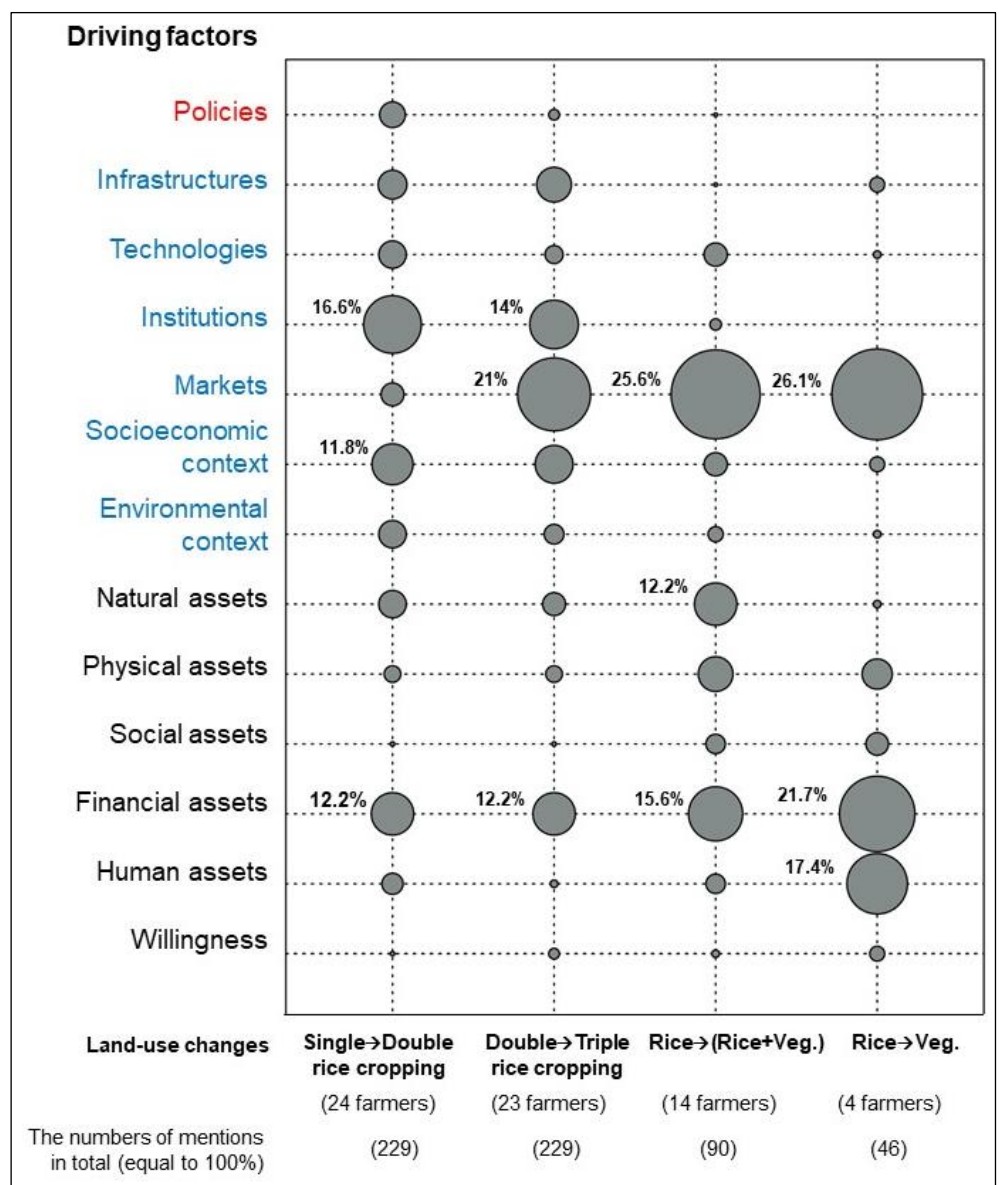

**Figure 4.** The multi-scale drivers of land-use change in the highly flooded zone of the Vietnamese Mekong Delta from 1975 to 2016. The sizes of the circles represent the relative influence of the drivers according to interviews with farmers (See Appendix A, Table A2).

Figure 4 depicts the frequency with which farmers mentioned each of the multi-scale drivers as affecting their land-use decisions in the highly flooded zone. Drivers that were mentioned by a higher percentage of farmers were considered to have a more active or significant role in farmers' decision-making processes.

Overall, we found that multiple drivers stimulated land-use changes. The influence of the drivers varied both across the studied communities (over space) and at the different points of change (over time). Farmers always mentioned the desire to earn greater profit from agricultural production as a motivating factor in their decision to try a new farming system. However, our analysis found that for farms following a rice intensification pathway, the effects of drivers at the regional scale were more significant than the financial assets of the household. The shift from single rice cropping to double rice cropping was driven predominantly by the activities of local institutions, households' prospering financial situations, the socioeconomic context, and infrastructure development. Most farmers emphasized the role of local officials in mobilizing them to switch from a single rice crop to double rice. Farmers were well aware that these officials were mandated to implement the food security policy of the central government. Their job was to disseminate the national policy to each household in the community, introducing the high-yielding rice varieties to farmers and organizing cooperative groups to dredge and expand the canal network. Indeed, the 'institutions' factor, at 16.6%, was ranked highest among all of the drivers for the switch from single to double rice. Another important driver in the expansion of double rice was changes in socioeconomic context, such as the neighborhood effect and opinions of the community (categorized as socioeconomic context). The change from single rice to double rice did not happen simultaneously. Instead, double rice gradually spread throughout communities, with farmers observing the success of neighbors who had already switched. The second rice crop required that sufficient water be available in the dry season. Therefore, farms located near rivers and canals became pioneers in double rice cropping. Their prosperous and successful second harvests offered a strong motivation for neighbors to increase their number of rice crops per year. As the majority of farmers began to practice double rice cropping, water usage changed within the community. Since the VMD has a flat topography, if one field was inundated to irrigate paddy rice in the dry season, the neighbor's field was also affected. A neighboring farmer could then no longer continue to cultivate one rice crop in the rainy season followed by cash crops or vegetables in the dry season. Rather, they had to follow their neighbor and plant a second rice crop.

Drivers such as institutions, infrastructures, socioeconomic context, and financial assets played a substantial role in the shift from double rice to triple rice, as well. Here, however, farmers also mentioned various market-related factors which facilitated and motivated their decision to further intensify rice cropping. Indeed, the significance of 'markets' as a driver of land-use change rose from 6.6%, when farmers switched to double rice, to 21% when they switched to triple rice. Markets were the most frequently mentioned driver for the latter switch, followed by 'institutions', at 14% (Figure 4). The transition to triple rice required the construction of a high dike enclosure to prevent rice fields from being flooded in the rainy season. The only exception was in some remote regions where the annual floods came later, therefore allowing farmers to cultivate and harvest a third rice crop prior to the onset of flooding without the need for a high dike. This was the case Vinh Phuoc, according to farmers there. These farmers also reported that high dike construction required a government investment, and had to be approved by 60% of the farmers in the affected community. In our study area, it took up to ten years to gather the needed consensus. In these communities, prevailing opinion had a double-edged effect, delaying the implementation of the central government policy as well as forcing farmers to follow the majority's decision. The earliest high dikes in the study area were built in the western commune of Vinh Nhuan in the year 2000, with dikes in the other districts following in 2009, 2011, and 2012. A Vinh Nhuan farmer noted that high dike construction began after the historical flood of 2000. Moreover, the district was said to have already had relatively high traditional dikes, which the government considered comparatively easy and cost-effective to upgrade. Nonetheless, the construction of high dikes and canals here raised objections at first. Farmers had to contribute to dike construction costs in an amount based on their land size, to be paid in annual installments over three years—though they could not yet be sure of the success of a potential third cropping season. Dike construction

also meant that farmers had to give up some of their lands to accommodate the dike. They had to adapt their livelihoods, as well, particularly those, such as fishers, whose livelihoods depended on the floods. After ten years of successful earnings from double rice, as well as seeing neighbors benefit from triple rice, farmers did eventually become confident that they could earn more with a third rice cropping season. Many farmers also reported a need for increased earnings at this time, for example, to pay school tuition fees for children. Others, particularly older farmers, said they were interested in giving up secondary jobs. By increasing their number of rice crops, they expected to be able to earn more income from the farm without too much additional labor. Rice cultivation also received abundant agricultural services, in addition to support provided to rice markets, commodity agents, and rice companies. In sum, the transition to triple rice in the VMD was driven by government policies, dike construction, and neighborhood effects, but markets were the driver with the greatest impact.

In comparison to a rice monocrop, the mixed farming system of rice and vegetables occupied a very small area of the highly flooded zone. However, many farmers expressed a preference for the cultivation of both rice and vegetables if the conditions were feasible. We spoke with few farmers who indicated wanting to farm only vegetables. Here, again the market element played a major role in farmers' desire to diversify their products. 'Markets' were mentioned as a motivator by some 26% of farmers who had switched to a mixed farming system. Other farm-level drivers for a switch to the mixed farming system were, in decreasing order of importance, financial assets, human assets, and the physical and natural assets of the farm. Our analysis also suggests that only farmers with strong financial and labor capacity could embark on a specialization in vegetables. Interestingly, farmers rarely mentioned the role of higher-scale factors, such as national policies, infrastructure, and local institutions, as considerations with regard to an eventual switch to vegetable farming (Figure 4). Rather, farmers who had considered switching to vegetables said they were attracted by the higher selling price for vegetables compared to rice. A successful season of vegetables, they said, could bring in much more income than intensive rice farming. However, vegetable cultivation was a capital-intensive operation with a high risk of crop and market failure. And vegetable production was made all the riskier by the perishable nature of the freshly harvested product. Vegetable cultivation also required long working hours and intensive labor. Therefore, diversification into vegetables occurred only on a small scale—1–2 ha per farm—in areas with favorable geographies. Vegetable cultivation was also difficult to expand and maintain, due to labor limitations (due to the high rate of outmigration) and a lack of storage facilities and post-harvest services. Moreover, output markets were unstable, with strongly fluctuating prices.

*4.3. Spatial and Temporal Aspects of Drivers*

In general, land uses in the five communes followed the rice-based intensification pathway. However, the changes did not occur simultaneously across all communes. Farmers in the western communes (Vinh Phuoc and Tan Lap) embarked on triple rice cropping later than those in the eastern communes (Vinh Nhuan, Vinh Thanh, and Binh Chanh). Within the communes, however, the transitions occurred at the same time. Since the studied communes were spread evenly from west to east across the highly flooded zone, our data enabled us to explore whether the drivers of such change differed over space and time.

Figure 5 shows the frequency with which the different drivers were mentioned in each commune. Overall, each commune reflected the general trend. Land-use changes were driven by multiple factors. For intensive rice farming, regional scale drivers played the most important role. Meanwhile, agricultural diversification was driven mainly by markets and farm-level factors. In the more western communes, the key drivers played a relatively equal role in the switch from single rice to double and triple rice cropping, especially in Vinh Phuoc (the most western commune). Differences were more clearly distinguishable for the eastern communes, where local institutions, national policies, socioeconomic context (e.g., opinions within the community), and markets were the more prominent drivers.

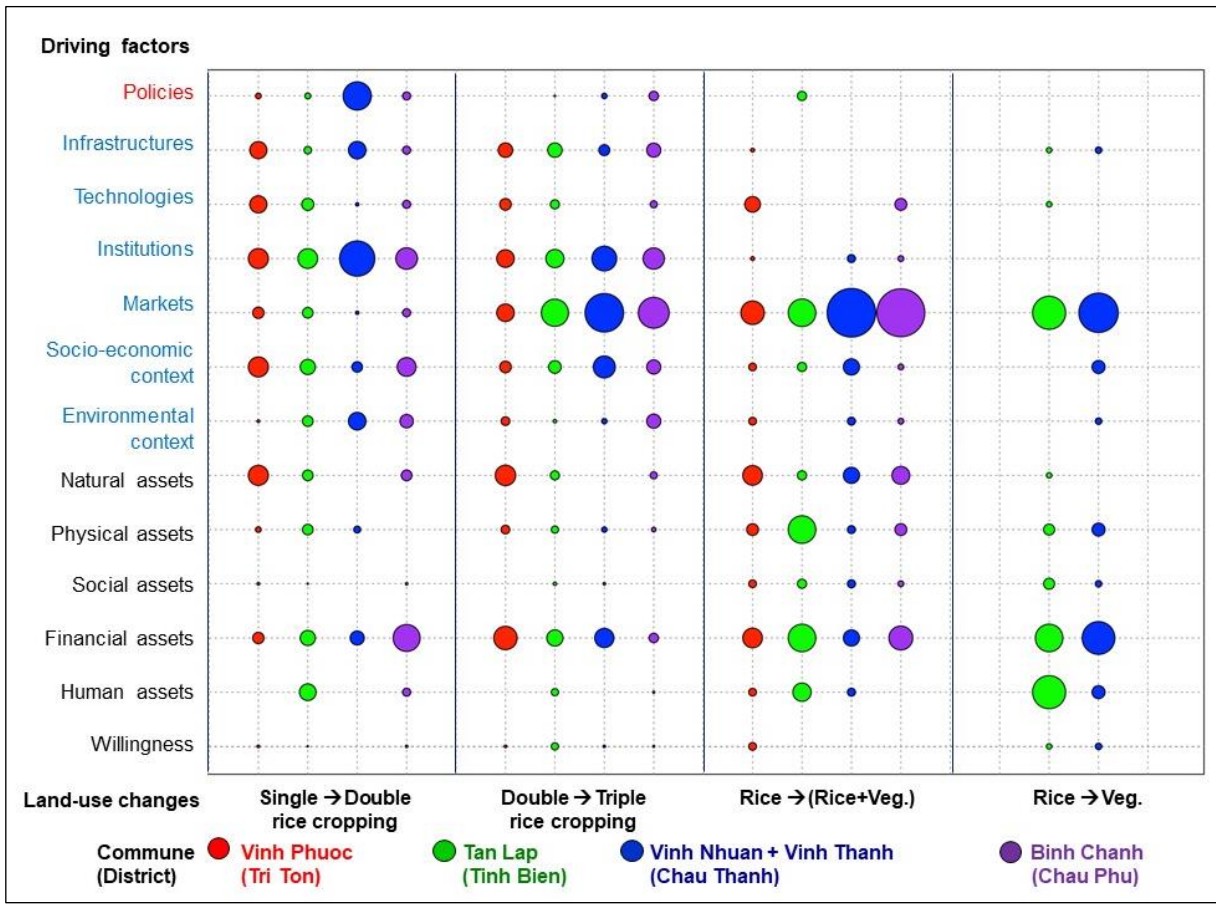

**Figure 5.** Influence of drivers of land-use changes in the five studied communes from 1975 to 2016. The sizes of the circles represent the frequency with which interviewed farmers mentioned the driver (See Appendix A, Table A3). Red and green circles represent communes located in the more western part of the study area. Blue and purple circles represent communes in the more eastern areas.

Output prices, natural and physical assets of the farm, and household financial assets were frequently mentioned by farmers when they spoke about the switch to a mixed farming system of rice and vegetables. However, the influence of these drivers was about equal. Meanwhile, in the more eastern communes, markets played the main role, mentioned by about 35% of these farmers as the reason for their switch to a mixed farming system. On vegetable farms, markets, financial assets, and human assets (labor) were the main drivers in both the west and east. Interestingly, the interviews with farmers brought a phenomenon of in-migration to light; that is, people were migrating from east to west to buy or rent larger and more fertile lands there, as cultivation in the west was less intensive. These migrants brought their experiences and ideas about farming with them, thus potentially modeling successful farming systems that could become examples for neighbors to follow.

## 5. Discussion

For forty years, rice-based intensification has formed the dominant land-use change pathway in the highly flooded zone of the VMD. In terms of drivers, our findings align with those of previous studies, which show water management infrastructures as having a key role in the promotion of rice intensification in the upper floodplains [2,3,5,58]. Our findings also confirm the major role of national food security policies in precipitating rice intensification in the highly flooded zone, although national policies did not figure prominently in farmers' own accounts of their reasons for switching to more intensive rice cropping. Indeed, the large-scale reclamation of acid sulfate soils in the early 1980s and the vast expansion of high dikes would have been impossible without central government

support and financial investments [49]. Our research also identified a growing impact of markets in driving land-use changes in the VMD, reflecting shifting national policies. In 1986, *Doi Moi* brought the decollectivization of land and agricultural production in the delta [18], while liberalization policies after 1989 [50] enabled markets to develop and allowed farmers to engage with them directly. The role of markets as a driver of change was even more explicit in the salinity intrusion zone. Our companion article on that case further analyzes and discusses the varying roles of multiple drivers in different socio-hydrological settings [42].

The case study presented here, in the highly flooded zone of the VMD, underscores the value of examining the diverse, multi-scale drivers of land-use changes. We found that each driver had an independent role in the process of land-use change, particularly at the farm level. However, the drivers influenced each other as well, accordingly changing over time. Conducting our study at the farm level enabled us to identify key drivers, such as local authorities and the community, both of which had a substantial impact on farmers' land-use decisions. Local institutions can be considered a downscaled version of central government policies, as local authorities in Vietnam are mandated to strictly enforce centrally issued decisions and resolutions [48]. The influence of the community, too, as a driver related to the socioeconomic context, stood out in the studied period. Farmers mentioned other drivers, too, which warrant greater attention in future research. For example, while many of the interviewed farmers were reaching advanced ages, young people were increasingly migrating to cities to work in non-agricultural jobs. Thong (2019) [58] found that while the development of high dikes and agricultural modernization supported rice-based intensification, it also forced the poor to migrate away from rural areas. This led to labor scarcity, which increased the cost of maintaining a household farm or of changing to a new farming system. The absence of agricultural heirs could result in lands being sold or leased to farmers migrating in from elsewhere. Changes in community demographics and the influence of migrants could set in motion drastic changes in regional land uses. In this regard, Hettig et al. (2016) [41] found that migrants may use unsuitable farming practices due to their limited knowledge of the local agroecology.

We found that the transition to more intensive rice cultivation usually occurred several years earlier in the eastern region of the study area compared to the western region. This variation in the timing of the change can be explained by proximity to the main rivers and the early building of water management infrastructures in the eastern region. Rice intensification, too, took place many years earlier in the freshwater alluvial zone in eastern An Giang Province (Figure 2). This can again be explained by physical or geographic drivers, as these inter-river plains were relatively higher in altitude, so they required less effort for dike construction to achieve sufficient flood protection [23]. At the time of our study, many farms in the freshwater alluvial zone had shifted from triple rice to vegetable cultivation or fruit production. Having been completely free from the effects of annual floods for more than three decades was an advantage that allowed farmers here to fully invest in agricultural diversification [59]. The question now is whether farms in the highly flooded zone can follow suit, and switch from intensive rice farming to specialized vegetable or fruit production in the near future. Our conversations with farmers in the highly flooded zone suggested that agricultural diversification for them will be more challenging, because the average farm size here was larger than in the freshwater alluvial zone (Table 1). Moreover, vegetable production is more capital intensive and riskier, with the risks associated with fruit orchards even greater. Most farmers living inside high dike enclosures said they hoped to continue their rice cultivation for years to come, especially triple rice cropping. Farmers considered intensive rice to be stable and the least risky land use—though they did recognize that it was not highly profitable or sustainable for their households. Enclosing lands with high dikes and the subsequent intensive land exploitation caused soil degradation, rodent infestations, and pollution in the study area. Tran et al. (2018) [9] found that after a while triple-rice cultivation households in the upper VMD floodplains did not earn higher profits than double-rice

households, due to the cost of the fertilizers and pesticides needed for the third rice crop. Farmers living outside the dike rings had more options, as the open water and sediment conditions allowed them to try alternatives, such as flood-based farming, which often yielded greater profit without the negative environmental impacts [60,61]. Indeed, the undesirable environmental consequences have prompted the Vietnamese government to reconsider its policy of prioritizing rice production and continued dike expansion [17].

In terms of methodology, the empirical basis of our research did yield a deeper understanding of the role of multiple drivers at each juncture in land-use change pathways in the VMD. Our interviews covered a particularly long period in the history of land-use change in the VMD (40 years) through the experiences and perspectives of 31 local farmers. A literature review on land-use change in the highly flooded zone was carefully undertaken before the interviews [3,5,55,62]. Studied districts were selected with consideration to cover the regional heterogeneity. Thanks to recommendations by local agricultural officials, we were able to interview farmers with all types of land uses that existed in the four communes. Our methodology also enabled us to confirm and intensively interrogate the findings from our interviews, as almost half of our participants were interviewed twice, with additional questions being asked. Thus, our conclusion is sufficiently relevant to represent the population in the highly flooded zone despite the small number of interviews.

We did not present farmers' accounts of land-use changes in the typical descriptive way, with quotes and anecdotes. In an effort to quantify qualitative data, we calculated the frequency with which particular change drivers were mentioned, using a matrix and color coding to count these frequencies. We then structured the findings using our multi-scale drivers framework. The systematic nature of this qualitative analysis enabled us to produce more concise and reliable interpretations. The framework allowed us to analyze land-use changes in multiple dimensions, including the timeline of changes, drivers of change, and spatial scales. For instance, the land-use change pattern was found to be uniform across the four studied districts, but the time of implementation differed per region and community, related to differences in the influences of the key drivers.

In addition, it must be noted that transcript analysis, such as the one conducted in the current research, is a very intensive and time-consuming process. It furthermore requires the analyst to have a good grasp of the context of the interviews, as well as the geographic and demographic characteristics of the study areas, and the local cultural setting. This may restrain the reproducibility of this research in a broader region, with plenty of interviewed groups. We therefore recommend coarse-scale studies to employ alternative data-collecting approaches to reduce processing time. For example, pathways of land-use change and critical driving factors can be investigated through a literature review, key informant interviews, or focus group discussions. Then, questionnaire surveys with numerous samples can be conducted to calculate the influence value of driving factors. These alternative approaches are often used in recent studies [9,21,60,61,63–65] thanks to their workable timing and the diverse information gained. However, we purposely chose the semi-structured interviews and transcript analysis for this research because they allowed us to explore the specific context behind each land-use change pathway of an individual household, as well as to deeply understand the attitude and perspective of farmers. We conducted a similar case study in the coastal salinity intrusion zone to test the validity and rigor of the developed framework. That case study, alongside further analysis and a cross-case comparison, will be presented in a companion paper [42].

## 6. Conclusions

Our research investigated 40 years of land-use changes in the Long Xuyen Quadrangle floodplain of the VMD. The developed multi-scale drivers framework enabled us to conduct both qualitative and quantitative analyses of the multiple drivers that farmers mentioned as influencing their decisions to change land uses. We calculated the contribution of each driver by the frequency with which it was mentioned by the interviewed farmers, and farmers' narratives were used to elaborate key relations. We found drastic changes in land

uses, with the predominant path being toward more intensive rice cultivation. Among the 40 drivers of change mentioned by the interviewed farmers, the drivers operating at the regional scale played a particularly large role. The expansion of canal networks and the construction of high dikes were prerequisites for rice intensification. However, farmers' individual decisions on land-use changes were driven largely by local institutions and prevailing opinions within the community. The role of markets as a driver of land-use change grew over time. The national policy prioritizing food security and the *Doi Moi* reforms brought great opportunities for rice production. Rice intensification received strong support from favorable rice export markets, as well as from the available agricultural services. More recently, market forces played an important role in motivating farmers to try vegetable farming, in addition to farms' specific physical assets (geography), financial assets (capital), and human assets (labor capacities within the household). However, at the time of this research, the shift from intensive rice to agricultural diversification had not gained momentum. This was due to the upper delta's agro-hydrological context, which still favored triple rice cropping. This is a challenge that will have to be addressed from higher levels, for example, through a regional support system and stable markets for diverse agricultural products, incentives to increase the labor pool, and the dissemination of innovative technologies.

The role of market-oriented drivers emerged even more clearly in our second case, in the VMD coastal area where freshwater resources were threatened by salinity intrusion. This is addressed in a cross-case analysis in a companion article [42].

**Author Contributions:** Conceptualization, Methodology, Investigation, Formal analysis, Visualization, Data Curation, Writing—Original Draft, T.N.L.; Supervision, Conceptualization, Methodology, A.K.B.; Supervision, Conceptualization, Writing—Reviewing and Editing, G.E.v.H.; Supervision, Conceptualization, Project administration, P.J.G.J.H.; Methodology, T.T.T.N. All authors have read and agreed to the published version of the manuscript.

**Funding:** This research was funded by the Dutch organisation for internationalisation in education (Nuffic), under the program of The Netherlands Initiative for Capacity development in Higher Education (NICHE), grant number: Nuffic/Niche/104/VNM. This is a collaboration project between Wageningen University and Research (WUR), the Netherlands, and the Centre of Water Resources Management and Climate Change (WACC), Institute for Environment and Resources, Vietnam National University-Ho Chi Minh City (VNU-HCM), Vietnam.

**Data Availability Statement:** Not available.

**Acknowledgments:** We extend special thanks to Long Hoang Phi, postdoc researcher in the Water System and Global Change Group, WUR, for developing an R-code to illustrate our findings (Figures 4 and 5), as well as for his thoughtful comments to sharpen the 'Results' section.

**Conflicts of Interest:** The authors declare that they have no known competing financial interests or personal relationships that could have appeared to influence the work reported in this paper.

## Appendix A

**Table A1.** List of driving factors mentioned by farmers in An Giang province, Vietnamese Mekong Delta.

| Driving Factors | Sub-Driving Factors | Explanations Based on the Interviews |
|---|---|---|
| (A) Policies | A1. The collectivization policies (1975–1986) | - Farmers worked in cooperative groups which were organized by the government.<br>- Free market was not allowed. The government bought and allocated all products. |
| | A2. The re-arranging land ownership policies (1975–1986) | - Farmers had to hand over their lands to co-operative groups. Then, the government allocated land to each household, and the size of the land was decided according to the number of members in each household.<br>- The government allocated lands to (poor) immigrated farmers who participated in the "New Economic" policy.<br>- Farmers who owned (large) land but lived in other communes/districts had to give up their land and return to their hometown.<br>- When the economic collectivization policies failed, the government dissolved the cooperative groups. Derelict lands were sold to farmers. The ex-owners of those lands (who had returned to their hometown) were allowed to come back to redeem their lands. |
| | A3. Food security policies (from 1986 to present) | - The policies aimed to increase the annual rice production to feed the people, to export, and to earn more profit from rice farming (focusing on quantitative production). Farmers were therefore encouraged to intensify their cropping seasons.<br>- The state government controls the price of rice production and supports the government's rice-companies by policies.<br>- The state government invested in new infrastructures and agricultural services to support the intensive rice farming. |
| | A4. Natural disaster prevention and control policies | - The state government built and enhanced infrastructures such as concrete dikes and elevated roads to protect lives and properties of the people from flood hazards.<br>- The government's army participated in public works to help the people when disaster happened (severe floods, broken dikes). |
| | A5. "New Rural" movement (since 2010) and "Large rice field" program (since 2013) | - The government invested to build concrete roads and bridges in the rural areas.<br>- The government encouraged farmers to build concrete houses, or to make the base of their house equal to the height of high dikes.<br>- The government encouraged farmers to participate in the "Large rice field" program. |

Table A1. *Cont.*

| Driving Factors | Sub-Driving Factors | Explanations Based on the Interviews |
|---|---|---|
| (B) Infrastructures | B1. Canals (water supply and drainage system) | - The government operated public works to dredge and enlarge the existing canal networks, aiming to (1) release flooding water faster, (2) provide more fresh-water for irrigation, and (3) reclaim lands from sulfate acid soils.<br>- The government dredged new canals in the remote areas. |
| | B2. Low dykes/August dykes | - The local government and farmers worked together to build and maintain the low dikes to delay flooding times, and farmers could harvest their rice. |
| | B3. High dykes/full dykes (also function as concrete roads) | - The government conducted projects to build new high dikes, or to elevate the existing low dike systems to completely protect the areas from flood hazards.<br>- The government built new national and regional roads which also function as high dikes |
| | B4. Others (bridges, electricity, domestic/fresh water supply, etc.) | - The government improved the small roads and built concrete bridges.<br>- The government improved the fresh-water supply system for people in remote areas.<br>- Farmers extracted ground-water or stored rain-water for domestic usages.<br>- Farmers bought the solar electricity equipment to generate electricity for pumping and daily usages. |
| (C) Technologies | C1. Farmers improved the physical conditions of their farms. | - Farmers used fresh water to treat sulfate acid soils.<br>- Farmers used pumps to drain flooding water, or to irrigate the crops.<br>- Farmers used fertilizer, agrochemicals.<br>- Farmers changed to new rice/vegetables varieties that were more adaptable to their farming conditions<br>- Farmers applied new techniques to improve the physical conditions of their farms. |
| | C2. Financial and technical support from the government | - Farmers tried the new rice varieties introduced by the government.<br>- Farmers participated in technical training and tried the high value farming models introduced by the local government.<br>- If there was a funded program to try new farming models, the farmers who participated would receive financial and technical support. |
| | C3. Farmers participate in high value farming models | -Farmers participated in technical training and tried the high value farming models introduced by the private sector or their friends/family/etc. |

**Table A1.** *Cont.*

| Driving Factors | Sub-Driving Factors | Explanations Based on the Interviews |
|---|---|---|
| (D) Institutions | D1. Co-operative groups | - After the war, farmers had to hand over their lands to co-operative groups which were operated by the government. They worked together in the co-operative groups to produce rice and participate in public works (dredging canals, building dikes, pumping . . . )<br>- Nowadays, farmers do not have to work in the co-operative groups anymore, but this kind of group still exists and is in charge of pumping and some agricultural services. |
| | D2. Local officers gathered the consensus of farmers | - Local officers visited farmers, or invited them to the community meeting to introduce policies.<br>- Local officers organized frequent meetings, trying to persuade farmers to follow the state government's policies as much as possible.<br>- Local officers gathered the votes of the majority. When the government obtained the major consensus of farmers, they could conduct a project in the commune. |
| | D3. Local officers operated public works and developed infrastructures | - The local government received funding from the state government to invest in building new high dikes, canals, and roads.<br>- The local government is in charge of maintaining the canals and dikes, but the farmers need to pay fees for those public works. |
| | D4. Financial and technical support program launched by the government | -The local government introduced the high yield rice varieties to farmers.<br>- The local government looked for high value farming models that met the government's plans, and offered farmers technical training.<br>- Agricultural officers introduced farmers to the rice companies or funded projects. |
| | D5. Introducing farmers to high value farming models | Local authorities cooperate with companies and organizations to introduce high value farming models to farmers. |

**Table A1.** *Cont.*

| Driving Factors | Sub-Driving Factors | Explanations Based on the Interviews |
|---|---|---|
| (E) Markets | E1. Agricultural services | - Farmers paid fees to build/maintain dikes and canals, or to pump water in the rainy season.<br>- Conditions to rent agricultural machines and laborers.<br>- Conditions for transportation and storage. |
| | E2. Traders and markets | - Trends and demands of agricultural products in the market.<br>- The confidence between farmers and traders. For example, traders can ask farmers to grow a certain type of rice/vegetables, or they can sign a contract with secured/fixed price before the start of a cropping season. However, sometimes traders do not follow the contract, they do not come back to buy the products or they only buy if farmers reduce price because the price in market has dropped dramatically.<br>- Conditions to do business with traders. For instance, traders preferred to buy products from the major farmers who produced the same type of crops. Those farmers had more advantages in transporting, storing, and selling than the minor farmers.<br>- Conditions to do business with agricultural agents or agricultural extensions. For example, farmers can pay for input products (e.g., fertilizers, agrochemicals, varieties, etc.) after finishing the harvest. |
| | E3. Price of agricultural products | - The fluctuation of prices in market<br>- The differences in prices among agriculture/aquaculture products (e.g., vegetables vs. rice, chili vs. watermelon, fishes vs. rice/vegetables, etc.)<br>- The differences in prices when doing business with the rice companies (secured/fixed prices), in comparing with the traders/open markets (fluctuated prices). |
| | E4. Rice companies | - The companies offered a secured price, technical training, and other supporting services for farmers if they signed a contract to sell rice to the companies (mostly the state's companies). |
| | E5. Household expenses | - Everything nowadays must be bought from the market, even fish or vegetables; the living expense therefore has increased. That makes farmers want to earn more profit from their farming.<br><br>- The farmers want their children to study in college, hoping their next generation can find a non-farming job. To pay the school fees, farmers need to earn more profit from farming.<br>-The middle-aged farmers are not interested in secondary jobs. By increasing the amount of rice cropping, they expected to earn more income without expending too much labor.<br>- Economic condition of the household are stable, farmers feel satisfied with their farming. They do not want to invest in a new farming system. |

**Table A1.** *Cont.*

| Driving Factors | Sub-Driving Factors | Explanations Based on the Interviews |
|---|---|---|
| (F) Social-economic context | F1. Opinion of majority group/pressure from the community | - For decisions related to policy, an individual farm has to follow the major group in the community (the opinions of 60% of farmers). <br> - The farmers who own large lands or succeed in farming have influences on others' opinions. |
| | F2. Neighborhood effects/success or failure of someone in the community | - When farmers see the success of others (neighbors, relatives, other communes, etc.), they are motivated to change their farming system. <br> - When many surrounding farms changed to a new land-use, a farm had to follow the neighbors because those farms influence each other in water usage, pest control, or selling products, etc. |
| | F3. Immigration | - Farmers move out or move back to the commune due to the changes in land-ownership policies. They farm in the commune but live in another commune/their original hometown. <br> - Farmers move to the commune because they inherit land from their parents, or when they get married (e.g., moving to the wife's hometown). <br> -Farmers living in the eastern region move to the western regions because land in the eastern region is becoming smaller (parents divide their land into smaller parts and give the land to their children), or because the lands have been overexploited. <br> - The poor farmers move to the remote areas where they can afford larger land. <br> -Farmers from other regions bring their experiences and ideas about a new type of land-use to the commune. |
| (G) Environmental context | G1. Local weather and flooding regime | - Changes in local weather (rainfall pattern, sunny days, temperature, etc.) affect all farms. <br> - Flooding regime is similar for the whole region if there is no high dike. |
| | G2. Agriculture zone (bounded by canals and dykes) | - Farms that have been bounded by the same dikes, canals, and roads share the same infrastructure, water, flooding conditions, etc. <br> - If the farms are in an agricultural zone, the farmers have to practice a certain farming system that was planned by the government (mostly at provincial level). For example, large scale vegetables or fish farms are not allowed in the triple rice zone. |
| | G3. Farming schedule | - Agricultural activities (transplant, pump water, harvest, spray agrochemicals, etc.) need to follow the same schedule. For example, a farmer cannot irrigate his rice field if his neighbors are going to harvest rice or vegetables. |

Table A1. *Cont.*

| Driving Factors | Sub-Driving Factors | Explanations Based on the Interviews |
|---|---|---|
| (H) Natural Assets | H1. Land form (high or low, flat or rough) | - Lands with low elevation and that are flat are easily inundated when it rains, and are not suitable for growing vegetables but can be used to grow rice.<br>- Lands with high elevation are feasible for growing vegetables<br>- Lands with rough surface require much labor to be feasible for agriculture. |
| | H2. Soil quality | - The concentration of sulfate acid in soils can be reduced or increased due to the available fresh water. Sulfate acid is more active in the dry season than in the rainy season.<br>- Quality of alluvial soils degrades due to less flooding time (less sediment), intensive use of agrochemicals, or over exploitation.<br>- Quality of alluvial soil can be improved due to fertilizers or crop-rotation. |
| | H3. Water availability | -Without canals, farmers do not have enough fresh water to treat the sulfate acid soil or to irrigate their crops. Their farming depends on rainwater. Thus, they only grow rice in the rainy season. In the dry season, they grow vegetables.<br>- Farmers open the sluice gate or pump fresh water from the canals to treat the sulfate acid soil and to irrigate their rice/vegetables.<br>- Too much water due to heavy rainfall will cause damage to productions. Thus, farmers have to pump water out of their fields.<br>-Being inundated by flooding can help to control pests and mice and to fertilize the fields. |
| (I) Physical Assets | I1. Location (close or far from roads, canals) | - The farms that are close to the high dike or roads have advantages in transportation, water-use, etc., but they had to give a part of their land when those infrastructures were built.<br>- The farms close to main and sub-canals are more active in using water resources.<br>-In the farms close to high dykes/roads, farmers can use the side part of the dykes/roads to grow vegetables because this part has high elevation and it is easy to transport vegetables.<br>- If farmers have many lands but they are not located in the same commune, farmers may prefer to work on the land near their house, and the remote one can be rented out. |
| | I2. Land size (large or small) | - Farmers who have small land (less than 0.5 ha) prefer high value farming systems such as vegetables and mix-farming, or may rent more lands to grow rice.<br>- Farmers who have large lands prefer the rice farming system because it requires less labor, the agricultural services are convenient, and the market fluctuation is less risky.<br>- Farmers who have many lands can try different land-use types, or keep their lands for rent. |
| | I3. Physical conditions were improved by farmers | - Sulfate acid soils were treated.<br>- Pumping flooding water from the fields.<br>- Fertilizer, agrochemicals, or organic materials were used to improve soils' quality |

**Table A1.** *Cont.*

| Driving Factors | Sub-Driving Factors | Explanations Based on the Interviews |
|---|---|---|
| (J) Social Assets | J1. Social network | - Farmers who also work for the local government or have a relationship with local officers can have advantages in updating information/policies, or have a bigger role in the community.<br>- Famers can have confidence/trust from the community.<br>- Famers can travel to many places to see and learn from other communes, districts.<br>- Farmers have close relationship with traders, companies, agricultural extensions, or some farmers are also traders, a service providers, or agriculture agents.<br>- Farmers can work together with their siblings, relatives, or cousins, or they can learn new things from them if their family's members live far away. |
| (K) Financial Assets | K1. Extra incomes/other livelihoods | - Farmers can have other livelihoods that contribute to the household income. When they travel to other regions to do the other job, they can observe and learn new farming systems.<br>- The children have jobs and send a part of their salary to their parents, and contribute to the household's income.<br>- By increasing the number of cropping seasons, farmers can have extra income. |
| | K2. Property or household's capital | - Economic status of the household (wealthy or poor, do they have savings in the bank).<br>- If the agricultural equipment belongs to the household farmer can use it or make some money by leasing it. Farmers may have to rent the equipment and share them with other farmers.<br>- Lands are property, so farmers can lease or mortgage their field to gain some income or capital.<br>- The ability to access loans from the bank and financial supports from the government.<br>- The ability to borrow money from their network (friends, relatives, agricultural agents, or mortgagees, etc.).<br>- To be in debt is a pressure for farmers. They have to extend and intensify their farming, such as by keeping doing triple rice cropping. It is risky for them to change to vegetables because it requires more investment while the market of vegetables is much more fluctuating. |
| | K3. Production or profit earned from farming system | - The difference (lower/higher) in production between two farming systems, leading to difference in profit. Farmers consider this difference before changing their farming system. They are usually motivated when seeing a chance to gain higher production and profit.<br>- Farmers prefer a farming system that has a stable production and profit (the production and the profit depend on both environmental conditions and market). |

**Table A1.** *Cont.*

| Driving Factors | Sub-Driving Factors | Explanations Based on the Interviews |
|---|---|---|
| (L) Human Assets | L1. Age | - Age and heath condition affects the practice of current farming system.<br>- Age affects the willingness to try new farming systems.<br>- Age affects the capacity to learn new farming techniques. |
| | L2. Education/knowledge/experience | - Farmers with reading ability can learn farming techniques from books and magazines.<br>- Farmers can learn farming knowledge from magazines, television, workshops, etc.<br>- Farmers who have many experiences in farming are confident in their current farming system. |
| | L3. Farming history/system memory | - Farmers keep farming system because it is a tradition of the family.<br>- Special events in the famers' life can make them change a new farming system (getting married and moving to the wife's family).<br>- Farmers who have tried many land-use types or livelihoods are more active and confident in changing their farming system. |
| | L4. Family structure/family labors | - The number of children in the family.<br>- Age, education level, and marriage status of the children.<br>- The number of people in the family who are using that farming system.<br>- The main labor source, who can decide which farming system is used for the household. |
| (M) Willingness | M1. Interests | - Farmers can prefer a certain farming system and want to keep doing that farming system.<br>- Farmers have self-motivation to try new farming systems.<br>- Farmers can prefer to follow the major farming system in the community. |

**Table A2.** Results: Driving factors mentioned by farmers in An Giang province, Vietnamese Mekong Delta.

| Driving Factors | Sub-Driving Factors | Single to Double Rice (24 Farmers) | | Double to Triple Rice (23 Farmers) | | To Rice + Vegetables (14 Farmers) | | To Only Vegetables (4 Farmers) | |
|---|---|---|---|---|---|---|---|---|---|
| | | Number of Mentions | % | Number of Mentions | % | Number of Mentions | % | Number of Mentions | % |
| (A) Policies | A1. The collectivization policies (1975–1986) | 2 | | 0 | | 1 | | 0 | |
| | A2. The re-arranging land ownership policies (1975–1986) | 5 | | 0 | | 0 | | 0 | |
| | A3. Food security policies (from 1986 to present) | 10 | 17 / 7.4 | 2 | 7 / 3.1 | 0 | 1 / 1.1 | 0 | 0 / 0 |
| | A4. Natural disaster prevention and control policies | 0 | | 4 | | 0 | | 0 | |
| | A5. "New Rural" movement (since 2010) and "Large rice field" program (since 2013) | 0 | | 1 | | 0 | | 0 | |

**Table A2.** *Cont.*

| Driving Factors | Sub-Driving Factors | Single to Double Rice (24 Farmers) | | Double to Triple Rice (23 Farmers) | | To Rice + Vegetables (14 Farmers) | | To Only Vegetables (4 Farmers) | |
|---|---|---|---|---|---|---|---|---|---|
| | | Number of Mentions | % | Number of Mentions | % | Number of Mentions | % | Number of Mentions | % |
| (B) Infrastructures | B1. Canals (water supply and drainage system) | 16 | | 4 | | 1 | | 0 | |
| | B2. Low dikes/August dikes | 1 | | 0 | | 0 | | 0 | |
| | B3. High dikes/full dikes (also function as concrete roads) | 1 | 19 | 18 | 23 | 0 | 1 | 2 | 2 |
| | | | 8.3 | | 10 | | 1.1 | | 4.3 |
| | B4. Others (bridges, electricity, domestic/fresh water supply, etc.) | 1 | | 1 | | 0 | | 0 | |
| (C) Technologies | C1. Farmers improved the physical conditions of their farms | 8 | | 7 | | 5 | | 1 | |
| | C2. Financial and technical support from the government | 10 | 18 | 5 | 12 | 0 | 6 | 0 | 1 |
| | | | 7.9 | | 5.2 | | 6.7 | | 2.2 |
| | C3. Farmers participate in high value farming models | 0 | | 0 | | 1 | | 0 | |
| (D) Institutions | D1. Co-operative groups | 1 | | 1 | | 0 | | 0 | |
| | D2. Local officers gathered the consensus of farmers | 18 | | 15 | | 2 | | 0 | |
| | D3. Local officers operated public works and developed infrastructures | 9 | 38 | 11 | 32 | 0 | 3 | 0 | 0 |
| | | | 16.6 | | 14 | | 3.3 | | 0 |
| | D4. Financial and technical support program launched by the government | 10 | | 5 | | 0 | | 0 | |
| | D5. Introduce farmers to high value farming models | 0 | | 0 | | 1 | | 0 | |
| (E) Markets | E1. Agricultural services | 8 | | 16 | | 4 | | 3 | |
| | E2. Traders and markets | 3 | | 8 | | 9 | | 4 | |
| | E3. Price of agricultural products | 0 | 15 | 4 | 48 | 7 | 23 | 4 | 12 |
| | | | 6.6 | | 21 | | 25.6 | | 26.1 |
| | E4. Rice companies | 0 | | 10 | | 0 | | 0 | |
| | E5. Household expenses | 4 | | 10 | | 3 | | 1 | |
| (F) Social-economic context | F1. Opinion of majority group/pressure from the community | 12 | | 16 | | 1 | | 0 | |
| | F2. Neighborhood effects/success or failure of someone in the community | 11 | 27 | 8 | 25 | 5 | 6 | 2 | 2 |
| | | | 11.8 | | 10.9 | | 6.7 | | 4.3 |
| | F3. Immigration | 4 | | 1 | | 0 | | 0 | |
| (G) Environmental context | G1. Local weather and flooding regime | 5 | | 4 | | 2 | | 1 | |
| | G2. Agriculture zone (bounded by canals and dikes) | 10 | 18 | 8 | 13 | 2 | 4 | 0 | 1 |
| | | | 7.9 | | 5.7 | | 4.4 | | 2.2 |
| | G3. Farming schedule | 3 | | 1 | | 0 | | 0 | |

**Table A2.** *Cont.*

| Driving Factors | Sub-Driving Factors | Single to Double Rice (24 Farmers) | | | Double to Triple Rice (23 Farmers) | | | To Rice + Vegetables (14 Farmers) | | | To Only Vegetables (4 Farmers) | | |
|---|---|---|---|---|---|---|---|---|---|---|---|---|---|
| | | Number of Mentions | % | | Number of Mentions | % | | Number of Mentions | % | | Number of Mentions | % | |
| (H) Natural assets | H1. Land form (high or low, flat or rough) | 2 | | | 2 | | | 7 | | | 0 | | |
| | H2. Soil quality | 11 | 18 | 7.9 | 9 | 15 | 6.6 | 3 | 11 | 12.2 | 1 | 1 | 2.2 |
| | H3. Water availability | 5 | | | 4 | | | 1 | | | 0 | | |
| (I) Physical assets | I1. Location (close or far from roads, canals) | 4 | | | 3 | | | 3 | | | 1 | | |
| | I2. Land size (large or small) | 7 | 11 | 4.8 | 8 | 11 | 4.8 | 6 | 9 | 10 | 3 | 4 | 8.7 |
| | I3. Physical conditions were improved by farmers | | | | | | | | | | | | |
| (J) Social assets | J1. Social network | 3 | 3 | 1.3 | 3 | 3 | 1.3 | 5 | 5 | 5.6 | 3 | 3 | 6.5 |
| (K) Financial assets | K1. Extra income/other livelihoods | 4 | | | 7 | | | 1 | | | 3 | | |
| | K2. Property or household's capital | 7 | 28 | 12.2 | 5 | 28 | 12.2 | 3 | 14 | 15.6 | 4 | 10 | 21.7 |
| | K3. Production or profit earned from farming system | 17 | | | 16 | | | 10 | | | 3 | | |
| (L) Human assets | L1. Age | 0 | | | 1 | | | 0 | | | 0 | | |
| | L2. Education/knowledge/experience | 4 | 14 | 6.1 | 1 | 5 | 2.2 | 3 | 5 | 5.6 | 3 | 8 | 17.4 |
| | L3. Farming history/system memory | 5 | | | 1 | | | 0 | | | 2 | | |
| | L4. Family structure/family labor | 5 | | | 2 | | | 2 | | | 3 | | |
| (M) Willingness | M1. Interests | 3 | 3 | 1.3 | 7 | 7 | 3.1 | 2 | 2 | 2.2 | 2 | 2 | 4.3 |

**Table A3.** Results: Driving factors mentioned by farmers in the five studied communes in An Giang province.

| Land-Use Changes | Single to Double Rice (24 Farmers) | | | | | | | | Double to Triple Rice (23 Farmers) | | | | | | | |
|---|---|---|---|---|---|---|---|---|---|---|---|---|---|---|---|---|
| ** Communes (farmers) | VP (4) | | TL (6) | | VN + VT (6) | | BC (8) | | VP (3) | | TL (5) | | VN + VT (7) | | BC (8) | |
| Driving factors | * N.M. | % | N.M. | % | N.M. | % | N.M. | % | N.M. | % | N.M. | % | N.M. | % | N.M. | % |
| (A) Policies | 2 | 4.2 | 4 | 4.4 | 8 | 20.5 | 3 | 5.9 | 0 | 0 | 1 | 1.3 | 2 | 4 | 4 | 6.9 |
| (B) Infrastructures | 6 | 12.5 | 5 | 5.5 | 5 | 12.8 | 3 | 5.9 | 5 | 10.6 | 8 | 10.5 | 4 | 8 | 6 | 10.3 |
| (C) Technologies | 6 | 12.5 | 8 | 8.8 | 1 | 2.6 | 3 | 5.9 | 4 | 8.5 | 5 | 6.6 | 0 | 0 | 3 | 5.2 |
| (D) Institutions | 7 | 14.6 | 13 | 14.3 | 10 | 25.6 | 8 | 15.7 | 6 | 12.8 | 10 | 13.2 | 9 | 18 | 9 | 15.5 |
| (E) Markets | 4 | 8.3 | 7 | 7.7 | 1 | 2.6 | 3 | 5.9 | 6 | 12.8 | 15 | 19.7 | 14 | 28 | 13 | 22.4 |
| (F) Social-economic context | 7 | 14.6 | 10 | 11 | 3 | 7.7 | 7 | 13.7 | 4 | 8.5 | 7 | 9.2 | 8 | 16 | 6 | 10.3 |
| (G) Environmental context | 1 | 2.1 | 7 | 7.7 | 5 | 12.8 | 5 | 9.8 | 3 | 6.4 | 2 | 2.6 | 2 | 4 | 6 | 10.3 |
| (H) Natural assets | 7 | 14.6 | 7 | 7.7 | 0 | 0 | 4 | 7.8 | 7 | 14.9 | 5 | 6.6 | 0 | 0 | 3 | 5.2 |
| (I) Physical assets | 2 | 4.2 | 7 | 7.7 | 2 | 5.1 | 0 | 0 | 3 | 6.4 | 4 | 5.3 | 2 | 4 | 2 | 3.4 |
| (J) Social assets | 1 | 2.1 | 1 | 1.1 | 0 | 0 | 1 | 2 | 0 | 0 | 2 | 2.6 | 1 | 2 | 0 | 0 |
| (K) Financial assets | 4 | 8.3 | 10 | 11 | 4 | 10.3 | 10 | 19.6 | 8 | 17 | 9 | 11.8 | 7 | 14 | 4 | 6.9 |
| (L) Human assets | 0 | 0 | 11 | 12.1 | 0 | 0 | 3 | 5.9 | 0 | 0 | 4 | 5.3 | 0 | 0 | 1 | 1.7 |
| (M) Willingness | 1 | 2.1 | 1 | 1.1 | 0 | 0 | 1 | 2 | 1 | 2.1 | 4 | 5.3 | 1 | 2 | 1 | 1.7 |

* N.M.: Number of Mentions. ** Communes: VP (Vinh Phuoc); TL (Tan Lap); VN + VT (Vinh Nhuan + Vinh Thanh); BC (Binh Chanh).

| Land-Use Changes | To Rice + Vegetables (14 Farmers) | | | | | | | | To Only Vegetables (4 Farmers) | | | | | | | |
|---|---|---|---|---|---|---|---|---|---|---|---|---|---|---|---|---|
| ** Communes (farmers) | VP (3) | | TL (3) | | VN + VT (4) | | BC (6) | | VP (0) | | TL (2) | | VN + VT (2) | | BC (0) | |
| Driving factors | * N.M. | % | N.M. | % | N.M. | % | N.M. | % | N.M. | % | N.M. | % | N.M. | % | N.M. | % |
| (A) Policies | 0 | 0 | 1 | 6.7 | 0 | 0 | 0 | 0 | 0 | 0 | 0 | 0 | 0 | 0 | 0 | 0 |
| (B) Infrastructures | 1 | 2.9 | 0 | 0 | 0 | 0 | 0 | 0 | 0 | 0 | 1 | 4 | 1 | 4.8 | 0 | 0 |
| (C) Technologies | 4 | 11.4 | 0 | 0 | 0 | 0 | 2 | 8.7 | 0 | 0 | 1 | 4 | 0 | 0 | 0 | 0 |
| (D) Institutions | 1 | 2.9 | 0 | 0 | 1 | 5.9 | 1 | 4.3 | 0 | 0 | 0 | 0 | 0 | 0 | 0 | 0 |
| (E) Markets | 6 | 17.1 | 3 | 20 | 6 | 35.3 | 8 | 34.8 | 0 | 0 | 6 | 24 | 6 | 28.6 | 0 | 0 |
| (F) Social-economic context | 2 | 5.7 | 1 | 6.7 | 2 | 11.8 | 1 | 4.3 | 0 | 0 | 0 | 0 | 2 | 9.5 | 0 | 0 |
| (G) Environmental context | 2 | 5.7 | 0 | 0 | 1 | 5.9 | 1 | 4.3 | 0 | 0 | 0 | 0 | 1 | 4.8 | 0 | 0 |
| (H) Natural assets | 5 | 14.3 | 1 | 6.7 | 2 | 11.8 | 3 | 13 | 0 | 0 | 1 | 4 | 0 | 0 | 0 | 0 |
| (I) Physical assets | 3 | 8.6 | 3 | 20 | 1 | 5.9 | 2 | 8.7 | 0 | 0 | 2 | 8 | 2 | 9.5 | 0 | 0 |
| (J) Social assets | 2 | 5.7 | 1 | 6.7 | 1 | 5.9 | 1 | 4.3 | 0 | 0 | 2 | 8 | 1 | 4.8 | 0 | 0 |
| (K) Financial assets | 5 | 14.3 | 3 | 20 | 2 | 11.8 | 4 | 17.4 | 0 | 0 | 5 | 20 | 5 | 23.8 | 0 | 0 |
| (L) Human assets | 2 | 5.7 | 2 | 13.3 | 1 | 5.9 | 0 | 0 | 0 | 0 | 6 | 24 | 2 | 9.5 | 0 | 0 |
| (M) Willingness | 2 | 5.7 | 0 | 0 | 0 | 0 | 0 | 0 | 0 | 0 | 1 | 4 | 1 | 4.8 | 0 | 0 |

* N.M.: Number of Mentions. ** Communes: VP (Vinh Phuoc); TL (Tan Lap); VN + VT (Vinh Nhuan + Vinh Thanh); BC (Binh Chanh).

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
