# Peer review of "Multi-Scale Drivers of Land-Use Changes at Farm Level I: Conceptual Framework and Application in the Highly Flooded Zone of the Vietnamese Mekong Delta"

_land, doi:10.3390/land12071273_

Round 1

Reviewer 1 Report

In this manuscript, a multi-scale framework is developed to identify the drivers of land-use changes in the Vietnamese Mekong Delta (VMD). Overall, the research is well-structured, and the key findings are reported and discussed in a logical way. Nevertheless, there are still some problems to be solved and clarified before the acceptance of this manuscript.

1. The majority of cited references were written in 2018 or earlier. It is highly recommended to cite more recent references published within the past five years to update Section 1.

2. Besides, the conclusions are drawn from the perspectives of 31 interviewed farmers, which may not represent the broader population of the study regions. The authors need to state or clarify this limitation as well.

In addition, a few minor points are:

3. Figure 3 needs further revision to improve its readability.

4. The term "farm geography" (Line 711) needs to be clearly defined.

5. The font style in Section 2 needs to be double-checked.

6. It is suggested to replace the phrase "real impact" (Line 757-758) with a more appropriate expression.

Minor editing of English language required.

Author Response

In this manuscript, a multi-scale framework is developed to identify the drivers of land-use changes in the Vietnamese Mekong Delta (VMD). Overall, the research is well-structured, and the key findings are reported and discussed in a logical way. Nevertheless, there are still some problems to be solved and clarified before the acceptance of this manuscript.

Thank you for your comment and suggestion.

1. The majority of cited references were written in 2018 or earlier. It is highly recommended to cite more recent references published within the past five years to update Section 1.

I updated Section 1 and Section 5 with new articles published in 2019 and 2021. The order number of these references are:

2023: [1] (Statistical data)

2021: [63],[65]

2020: [28]

2019: [13], [14], [47],[60],[61],[66]

2. Besides, the conclusions are drawn from the perspectives of 31 interviewed farmers, which may not represent the broader population of the study regions. The authors need to state or clarify this limitation as well.

We are also concerned about the number of interviews and how it can affect the conclusions. This issue has been addressed in the revised version (lines 846-858).

To improve the representativeness of our findings, we first conducted a literature review on land-use changes and their driving factors in the VMD. We also carefully selected the studied communes that can represent the heterogeneity of the highly flooded zone. Thanks to recommendations by local agricultural officials, we were able to interview farmers with all types of land uses that existed in the four communes. Their land-use pathways and factors that influenced their land-use decisions were deeply interviewed until the information was saturated. Hence, we are confident that our study can provide a satisfactory conclusion for the highly flooded zone in the VMD.

In addition, a few minor points are:

3. Figure 3 needs further revision to improve its readability.

I made Figure 3 larger to improve its readability (page 14).

4. The term "farm geography" (Line 711) needs to be clearly defined.

In the original version, I used the term “farm geography” to mention the natural and physical assets of the farm, including landform, soil quality, water availability, location, land size, and improved physical conditions. Details of these driving factors were present in the appendix (pages 29-30)
In the revised version, I replaced the phrase “farm geography” with “natural and physical assets of the farm” to clarify this driving factor (line 36 and line 737).

5. The font style in Section 2 needs to be double-checked.

I changed the font style in  Section 2 to match with other sections (lines 209-269).

6. It is suggested to replace the phrase "real impact" (Line 757-758) with a more appropriate expression.

I used “ a substantial impact” to replace the phrase “a very real impact”  (lines 783-784).

Minor editing of English language required.

The manuscript was proofread and edited by a professional English editing service before it was submitted to the Land Journal. Language editing has not been done for the revised version because I have just made some minor changes, such as removing some paragraphs and rewriting some sentences in Section 1, Section 2, and Section 5. If the reviewer notices some incorrect uses of English in this version, I will check the language again.

Reviewer 2 Report

“Decision 899 [20] and Resolution 120” (p. 2) It is better to give the reader a brief summary of the content rather than numbering.

The introduction is too much about the challenges and policies of the study area, making it difficult to pay attention to the description of the key scientific issues of the article. Therefore, the authors are encouraged to consider placing some descriptive text in the study area.

As mentioned above, the introduction section needs to be further condensed, there are currently many descriptions that are not relevant to the scientific question (The multi-scale drivers framework), and many contents such as remote sensing can be summarized in a single sentence instead of the current paragraph

Figure 1. Why technologies is affiliated with Environmental context.

3.2 Date collection. Whether sampling is representative of regional heterogeneity should also be analyzed

“Figure 5” a legend is necessary

As the authors mentioned the time-consuming and labor-intensive investigation, the reproducibility and applicability of the framework proposed in this study to a large region needs to be emphasized and discussed.

Minor editing of English language required

Author Response

“Decision 899 [20] and Resolution 120” (p. 2) It is better to give the reader a brief summary of the content rather than numbering.

When I rewrote the text about the policy in Section 1, I added a brief description of Resolution 120 to the text (lines 107-108).

The introduction is too much about the challenges and policies of the study area, making it difficult to pay attention to the description of the key scientific issues of the article. Therefore, the authors are encouraged to consider placing some descriptive text in the study area.

As mentioned above, the introduction section needs to be further condensed, there are currently many descriptions that are not relevant to the scientific question (The multi-scale drivers framework), and many contents such as remote sensing can be summarized in a single sentence instead of the current paragraph

I agree with the reviewer’s comment that section 1 in the manuscript paid much attention to the challenges and political implementation of the Vietnamese Mekong Delta. I have shortened some paragraphs in Section 1 regarding the land uses, challenges, policies, and research methods that are not relevant.

Figure 1. Why technologies is affiliated with Environmental context.

In Figure 1, the arrow’s direction represents how the driving factors impact each other. Thus, the figure shows climate change, infrastructures, and technologies have changed the environmental context at the regional scale (page 6). This is a finding after I analyzed the interviews with farmers.

3.2 Date collection. Whether sampling is representative of regional heterogeneity should also be analyzed

The data were collected in January and April 2016 (line 482-483)

In section 3.1, I explained the four districts were selected because they exhibit the variety of biophysical and socioeconomic characteristics found in the various floodplain regions of the delta (lines 432-454). The eastern districts (Chau Thanh and Chau Phu) are closer to the river; their topography is flatter;  the percentage of area protected by high dikes is higher; and thus, their population economic status is better than the western districts (Tri Ton and Tinh Bien). The land-use patterns in the four districts had more than three-quarters of the land under intensive rice cropping, representing land uses in the highly flooded zone (Central Long Xuyen Quadrangle).

Section 3.2 presented the interviews conducted in five communes within the four studied districts (lines 456-471). Each commune had a land-use pattern and history of dike construction considered representative of its district. The snowball method was used to identify interviewees. Starting with interview subjects recommended by local agricultural officials, the number of farmers interviewed was determined by when we reached saturation of information.

“Figure 5” a legend is necessary

I edited the legend in Figure 5. Below the diagram, you can find the legend that assigns which colors represent which commune/ district (page 20).

As the authors mentioned the time-consuming and labor-intensive investigation, the reproducibility and applicability of the framework proposed in this study to a large region needs to be emphasized and discussed.

In the revised version, I have discussed the challenges of conducting a similar study in a larger region and suggested alternative approaches to collect data for the multi-scale driver's framework (lines 876-890).

Minor editing of the English language required

The manuscript was proofread and edited by a professional English editing service before it was submitted to the Land Journal. Language editing has not been done for the revised version because I have just made some minor changes, such as removing some paragraphs and rewriting some sentences in Section 1, Section 2, and Section 5. If the reviewer notices some incorrect uses of English in this version, I will check the language again

Reviewer 3 Report

Dear Authors,

I like this paper a lot; I think the idea of objectively quantifying peoples responses through formal text analysis has revealed some interesting nuances. I do however, have a few minor comments;

1) lines 237-239 grounded theory is usually promoted as a way to develop theory or hypothesis from the texts, I don't object to what you have done but the paper might benefit from a bit more detail into how you reconciled the formal structure in Figure 1 with the "free text" analysis.  You mention that the text analysis was time consuming, that I can agree with, and possibly helps explain the attraction of questionaires over your approach (again you might add a line as to the advantages and disadvantages of the alternative  approaches.

2) I struggled a bit reconciling figure 4, with the values in the appendix. For example in the transition from single to double cropping rice "policy" (7.4%) is not annotated on the figure, but "markets" (6.6%) is annotated. I think a more consistent rule like the top 3 or 5 reasons for each transition might be visually clearer?

Author Response

Dear Authors,

I like this paper a lot; I think the idea of objectively quantifying peoples responses through formal text analysis has revealed some interesting nuances. I do however, have a few minor comments;

Thank you for your comments and suggestions.

1) lines 237-239 grounded theory is usually promoted as a way to develop theory or hypothesis from the texts, I don't object to what you have done but the paper might benefit from a bit more detail into how you reconciled the formal structure in Figure 1 with the "free text" analysis.

In the revised version, I added a paragraph to explain how to apply grounded theory methods to the text analysis (lines 252-264). 

 You mention that the text analysis was time consuming, that I can agree with, and possibly helps explain the attraction of questionaires over your approach (again you might add a line as to the advantages and disadvantages of the alternative  approaches.

Thank you for your suggestion. In the revised version, I mentioned the challenges of conducting a similar study in a larger region and suggested alternative approaches to collect data for the multi-scale driver's framework. I cannot discuss the advantages and disadvantages of the alternative approaches because I did not use them in our study. However, I mentioned why those alternative approaches are often used in other studies and why we purposely choose the semi-structured interviews and transcript analysis (lines 876-890).

2) I struggled a bit reconciling figure 4, with the values in the appendix. For example in the transition from single to double cropping rice "policy" (7.4%) is not annotated on the figure, but "markets" (6.6%) is annotated. I think a more consistent rule like the top 3 or 5 reasons for each transition might be visually clearer?

I edited the values in Figure 4 according to your suggestion, using the rule of top 3 values (page 16).

Round 2

Reviewer 2 Report

The problem I raised has been resolved